# Generalized Additive Models via Direct Optimization of Regularized Decision Stump Forests

**Magzhan Gabidolla** [1]   **Miguel Á. Carreira-Perpiñán** [1]

## Abstract

We explore ensembles of axis-aligned decision stumps, which can be viewed as a generalized additive model (GAM). In this model, stumps utilizing the same feature are grouped to form a shape function for that feature. Instead of relying on boosting or bagging, we employ alternating optimization to learn a fixed-size stump forest. We optimize the parameters of each stump exactly through enumeration, given the other stumps are fixed. For fixed stump splits, the leaf values are optimized jointly by solving a convex problem. To address the overfitting issue inherent in naive optimization of stump forests, we propose effective regularization techniques. Our regularized stump forests achieve accuracy comparable to state-of-the-art GAM methods while using fewer parameters. This work is the first to successfully learn stump forests without employing traditional ensembling techniques like bagging or boosting.

## 1. Introduction

Machine learning has become increasingly prevalent, with many widely used, accurate methods being black box in nature such as tree ensembles and neural networks. Simpler models such as decision trees and rules, are usually less accurate, but have the advantage of being intelligible and interpretable. They are especially important for mission-critical systems such as in healthcare where models must be trusted. The focus of our paper is on one particularly important class of interpretable methods, a generalized additive model (GAM), which learns (nonlinear) functions over each individual feature, and additively combines them: $F(\mathbf{x}) = f_1(x_1) + \cdots + f(x_D)$. Because these do not model feature interactions, they can be visualized by plotting each shape function $f_d(x_d)$, making them highly interpretable.

Recently, GAMs have seen renewed interest in the machine learning community. One effective approach to constructing GAMs is through a forest of decision stumps. Because a stump (depth-one, axis-aligned decision tree) uses only a single feature, those using the same feature $x_d$ can be grouped, and this forms a shape function $f_d(x_d)$ for that one feature $x_d$. As building blocks for GAMs, stumps have the advantage of being flexible in choosing the feature, and during learning they can adapt to the data distribution by using more of themselves for more complex shapes while choosing less the simpler ones.

Boosting and bagging are by far the most widely used methods for learning tree ensembles, including stump forests. A recent, state-of-the-art implementation of GAMs, Explainable Boosting Machine (EBM), is based on these techniques of boosting and bagging. However, unlike the majority of established methods in machine learning, these techniques do not optimize a desired objective function over a parametric model of fixed size. Because no explicit objective function nor a learning problem is defined, they are more difficult to be analyzed theoretically, and have no guarantees of optimality. The way tree ensembles are learned using boosting and bagging is in stark contrast to the vast majority of established models in machine learning such as SVMs and neural networks. These are learned by defining an optimization problem over parameters of a fixed-sized model, and using an algorithm (e.g. gradient descent) to optimize it. In this paper, we propose, for the first time, to learn stump forests using this established machine learning paradigm: we define an objective function over a fixed number of stumps and optimize all parameters iteratively from an initial point (random, or an existing forest, which makes it possible to retrain the forest at any time in the future, e.g. with updated data). Since these models are non-differentiable, gradient-based methods are not applicable; instead, we effectively employ alternating optimization. Despite their simplicity, optimized stump forests are highly susceptible to overfitting. We analyze this behavior, and propose effective regularization based on the complexity of resulting GAM shape functions. Our optimized, reg-

[1]Dept. of Computer Science and Engineering, University of California, Merced, USA. Correspondence to: Miguel Á. Carreira-Perpiñán <mcarreira-perpinan@ucmerced.edu>.

*Proceedings of the 42nd International Conference on Machine Learning*, Vancouver, Canada. PMLR 267, 2025. Copyright 2025 by the author(s).

ularized stump forests achieve state-of-the-art performance on multiple regression and classification benchmarks.

After discussing related work in section 2, we explain the alternating optimization algorithm for stump forests in section 3. We then analyze the overfitting problem and propose regularization methods in section 4. Our experiments, detailed in section 5, validate the proposed methods on regression and classification benchmarks, demonstrating improved results over existing state-of-the-art methods. Additionally, we highlight the inherent interpretability of a GAM model through a case study on car price prediction.

## 2. Related Work

Generalized additive models have a rich history in statistics (Hastie & Tibshirani, 1986). As shape functions, splines are used almost exclusively, and with many different variations, including cubic splines, penalized B-splines (Eilers & Marx, 1996), and thin plate regression splines (Wood, 2003). Backfitting (Breiman & Friedman, 1983; Hastie & Tibshirani, 1986), a form of alternating optimization, and penalized (iteratively reweighted) least squares are commonly used to learn these GAMs. Selecting the placement and number of knots (a point where two spline segments meet) has been an open problem, with many different methods proposed to handle this (Breiman, 1993; Ruppert, 2002; Eilers & Marx, 2010). In our approach, the placement of knots is determined on the fly as a subproduct of the optimization, where stumps choose an optimal feature/threshold pair. Apart from splines, non-parametric methods based on trend filtering have been explored as shape functions (Petersen et al., 2016; Sadhanala & Tibshirani, 2019). We refer the reader to the books (Hastie & Tibshirani, 1990; Wood, 2017) for a more comprehensive treatment of GAMs in statistics literature.

More recently, other models for GAMs have been explored in the machine learning community. Lou et al. (2012) performed experimental comparison of different algorithms and shape functions for GAMs, and found out that ensembles of trees based on boosting and bagging produce most accurate results. Follow-up work (Lou et al., 2013) extended this to model pairwise interactions. Agarwal et al. (2021) used neural networks with special activation functions to model each of the 1D shapes in GAMs. Radenovic et al. (2022) proposed to use a single shared basis neural network to model all the shape functions and further extended them to pairwise interactions. Ibrahim et al. (2023) used soft decision trees to model both univariate and bivariate shapes, and imposed structural sparsity constraints on the number of terms. Given the differentiability of neural networks and soft decision trees, end-to-end SGD-based optimization methods were used to learn all these GAMs. Liu et al. (2022) used the idea of bi-

narizing features based on all possible thresholds, and training a linear model with $\ell_0$-penalty on these transformed features to obtain piecewise constant GAMs. Their feature binarization can be viewed as creating all possible stumps, and selecting them with $\ell_0$-penalty. In our approach we control the number of stumps explicitly using an $\ell_0$-like constraint.

Bagging and boosting are by far the most widely used methods to learn tree ensembles. Bagging is based on the idea of variance reduction by aggregating multiple diverse models trained on different bootstrap samples (Breiman, 1996). Boosting adds base learners sequentially with each added tree trying to improve the overall ensemble accuracy (Freund & Schapire, 1997). This can be done more effectively using an approximate form of gradient descent in function space, gradient boosting (Friedman, 2001). Traditionally, bagged or boosted tree ensembles have used a suboptimal, greedy recursive partitioning algorithm to learn the individual trees, such as CART (Breiman et al., 1984), and used axis-aligned trees. More recently, tree ensembles have been constructed by using more powerful tree models (such as oblique trees) that are properly optimized, which can be done using the Tree Alternating Optimization (TAO) algorithm (Carreira-Perpiñán & Tavallali, 2018). This results in a forest of significantly improved accuracy but using fewer and smaller trees, for both classification and regression (Carreira-Perpiñán & Zharmagambetov, 2020; Zharmagambetov & Carreira-Perpiñán, 2020; Gabidolla & Carreira-Perpiñán, 2022; Gabidolla et al., 2022). AdaBoost, the first practical boosting algorithm, has been especially successful with decision stumps (Freund & Schapire, 1996; Viola & Jones, 2004). However, bagging and boosting, considered as learning methods for decision forests, operate very differently from what the majority of methods in machine learning do, namely to define a parametric model of fixed size and structure and to train it by optimizing a well-defined objective function over all its parameters.

Very few works exist that follow this common learning paradigm with tree ensembles. Sorokina et al. (2007) proposed a Grove, an additive model of a fixed small number of regression trees, where each tree is axis-aligned and is trained suboptimally using greedy recursive partitioning on the residuals of the other trees. Carreira-Perpiñán et al. (2023) consider fixed-size forests of general types of trees, such as axis-aligned or oblique, where a well-defined objective function is optimized over all the forest parameters. This is done via alternating optimization over each individual tree (Forest Alternating Optimization (FAO)), where each tree is itself trained using TAO. In addition, a step is applied to optimize all the leaves jointly (given that all the decision nodes' parameters are fixed), a technique first used by Friedman & Popescu (2008). However, both of these ap-

proaches suffer from overfitting, and they both ultimately resort to an ensembling mechanism of either bagging or averaging. In our approach we use simpler forests, based on stumps. We control the overfitting behavior with appropriate regularization terms without any ensembling techniques. The way we optimize the stump forest can be seen as a particular case of FAO, but we can train each stump directly, without the need for TAO.

## 3. Direct Optimization of Stump Forests

To keep notation simple we use regression to introduce the problem. Assume we have a training set of $N$ (instance, target) pairs: $\{(\mathbf{x}_n, y_n)\}_{n=1}^N \subset \mathbb{R}^D \times \mathbb{R}$. Let $s(\mathbf{x}; \boldsymbol{\theta})$: $\mathbb{R}^D \to \mathbb{R}$ be a decision stump with 4 learnable parameters: $\boldsymbol{\theta} = \{\phi, \tau, \mu^l, \mu^r\}$. $\phi \in \{1, \ldots, D\}$ is a feature index to split, $\tau \in \mathbb{R}$ is a threshold value, $\mu^l, \mu^r \in \mathbb{R}$ are the left and right leaf prediction values. The predictive function of a stump $s(\mathbf{x}; \boldsymbol{\theta})$ works by comparing $x_\phi$ with $\tau$, and outputting either $\mu^l$ if $x_\phi < \tau$, or $\mu^r$ otherwise:

$$s(\mathbf{x}; \boldsymbol{\theta}) = \begin{cases} \mu^l, & \text{if } x_\phi < \tau \\ \mu^r, & \text{if } x_\phi \geq \tau. \end{cases} \quad (1)$$

A stump forest $F(\mathbf{x}; \boldsymbol{\Theta})$ is defined as a sum of $T$ stumps: $F(\mathbf{x}; \boldsymbol{\Theta}) = \mu + \sum_{t=1}^T s(\mathbf{x}; \boldsymbol{\theta}_t)$, with $\mu \in \mathbb{R}$ being a bias term. Since each stump uses only one feature, we can regroup the stumps by feature indices, and obtain an additive model:

$$F(\mathbf{x}; \boldsymbol{\Theta}) = \mu + \sum_{t=1}^T s(\mathbf{x}; \boldsymbol{\theta}_t) = \mu + \sum_{d=1}^D \sum_{t: \phi_t = d} s(\mathbf{x}; \boldsymbol{\theta}_t)$$

$$= \mu + \sum_{d=1}^D f_d(x_d) \quad (2)$$

where a shape function $f_d(x_d)$ of $d$'s feature is a sum of stumps that use feature $d$: $f_d(x_d) = \sum_{t: \phi_t = d} s(\mathbf{x}; \boldsymbol{\theta}_t)$. Since we are dealing with constant leaf predictor stumps, the shape function $f_d(x_d)$ is also a piecewise constant function with piece intervals corresponding to the stump thresholds. Viewing $F(\mathbf{x}; \boldsymbol{\Theta})$ as both a stump forest and an additive model will be important for our final learning method. Let us now first consider the learning problem of stumps forests without any regularization. For regression, we minimize a squared error:

$$\min_{\boldsymbol{\Theta}} \frac{1}{2} \sum_{n=1}^N [y_n - F(\mathbf{x}_n; \boldsymbol{\Theta})]^2. \quad (3)$$

Unlike in forward stagewise additive modeling of boosting, where stumps are added greedily to the forest, we want to directly optimize problem (3). Because stumps define a non-differentiable function, gradient-based methods are

not applicable. However, we can effectively apply alternating optimization: the parameters of one stump, while keeping the others fixed, can be optimized exactly; the parameters of all the leaf values $\{\mu_t^l, \mu_t^r\}_{t=1}^T$, while keeping the splits $\{\phi_t, \tau_t\}_{t=1}^T$ fixed, can also be optimized exactly. More specifically, alternating optimization consists of the following steps:

**Individual stumps.** The optimization problem (3) over a given stump $s(\cdot; \boldsymbol{\theta}_t)$ when others are fixed is $\min_{\boldsymbol{\theta}_t} \frac{1}{2} \sum_{n=1}^N [y_n - \sum_{u \neq t} s(\mathbf{x}_n; \boldsymbol{\theta}_u) - s(\mathbf{x}_n; \boldsymbol{\theta}_t)]^2$. This is a standard regression problem over a stump but with targets corresponding to the residuals. It can be solved exactly through enumeration over each (feature, threshold) pair as in traditional decision tree algorithms.

**All leaf parameters.** Once the splits $\{\phi_t, \tau_t\}_{t=1}^T$ are fixed, the problem (3) simplifies to a linear regression on features corresponding to stump partitions. To see this, we can rewrite the predictive function of a stump using an indicator function: $s(\mathbf{x}; \boldsymbol{\theta}) = \mu^l I(x_\phi < \tau) + \mu^r I(x_\phi \geq \tau)$. With this notation, the objective function over all leaves is:

$$\min_{\mu, \{\mu_t^l, \mu_t^r\}_{t=1}^T} \frac{1}{2} \sum_{n=1}^N \left[ y_n - \sum_{t=1}^T \frac{\mu_t^l I(x_{n\phi_t} < \tau_t)}{+ \mu_t^r I(x_{n\phi_t} \geq \tau_t)} \right]^2.$$

Since in this step all the splits $\{\phi_t, \tau_t\}_{t=1}^T$ are fixed, the indicator functions $I(\cdot)$ are just constants multiplying the unknowns $\{\mu_t^l, \mu_t^r\}_{t=1}^T$ that appear linearly inside the squared error. And so, eq. 4 is a simple linear regression problem on features induced by the stump splits, and can also be solved exactly and efficiently.

Starting with initial stump forest parameters (e.g. random), we can repeatedly optimize each stump individually, and all the leaves jointly. Because each such optimization step exactly solves the problem over its subset of parameters, this guarantees a monotonic decrease of the error. And since the total number of possible stump splits are finite, and since the algorithm either decreases the error or leaves it unchanged, this guarantees a convergence in a finite number of steps (assuming there are no cycles between states). While there are potentially many ways to order these alternating optimization steps, such as performing the step over all leaves after each individual stump step, or even doing everything in random order, in our experiments we use the following simple approach: first optimizing all leaves jointly, then optimizing the individual stumps in a fixed order (from 1 to $T$), and repeating this until convergence.

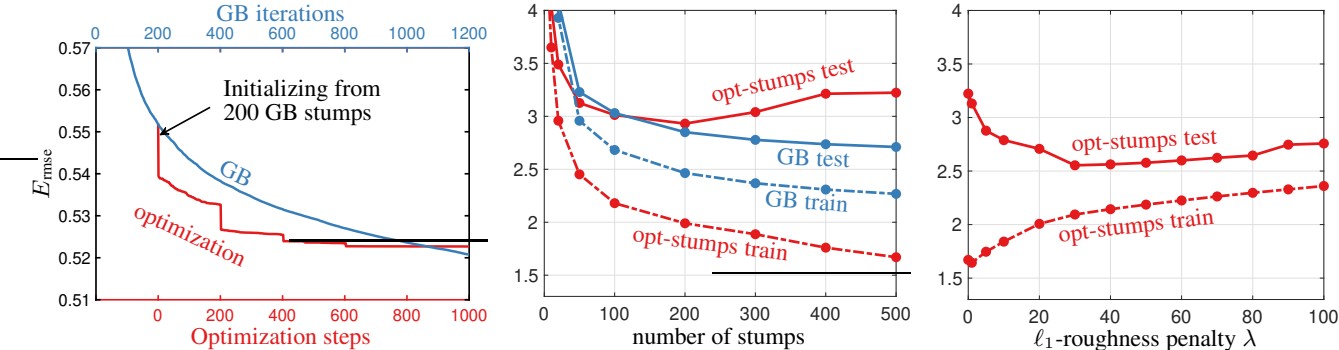

*Figure 1. Left:* Comparison of stump forest optimization against the greedy approach of gradient boosting (GB) in train error for the California Housing dataset. GB uses a learning rate of 1, i.e., no shrinkage. Optimization step corresponds to either the step over all leaves or over the individual stumps. *Middle:* Overfitting problem of optimized stump forests on the cpuact dataset. Here GB avoids overfitting by using a learning rate 0.3. *Right:* By penalizing the $\ell_1$-discontinuity we obtain stump forests with better generalization (cpuact dataset).

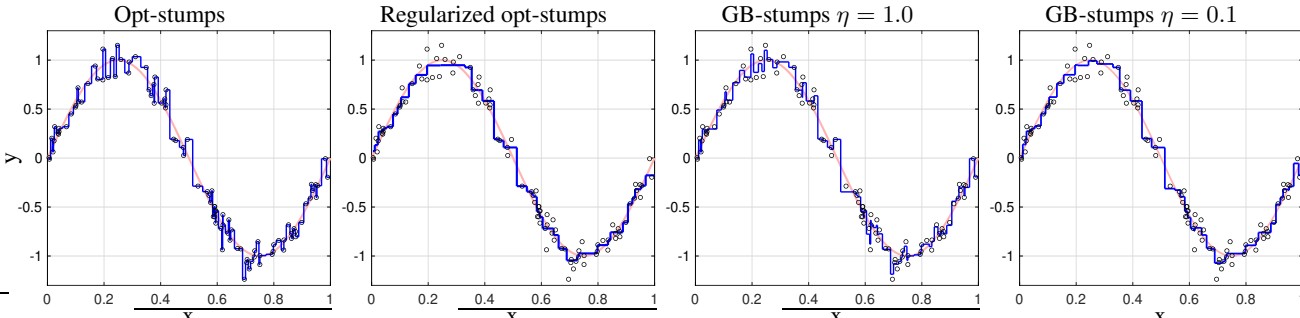

*Figure 2.* Illustration on a simple 1D regression problem. The learned shape functions are in blue. GB uses 1000 stumps with learning rates $\eta = 0.1$ and $\eta = 1.0$. Our optimization method uses 100 stumps. Ground truth targets are from a sine function plotted in light red.

## 4. Overfitting and Regularization

The alternating optimization method just described for stump forests is very effective at minimizing the objective function. The left plot of fig. 1 shows how it is possible to take 200 greedily added decision stumps (in a gradient boosting (GB) way with a learning rate of 1), and optimize it significantly with the proposed algorithm so that it matches the performance of more than 1000 GB stumps (the red vs the blue curves in the plot). Because GB works by greedily adding stumps, without revisiting the previous parameters, it produces much suboptimal results. In the left plot of fig. 1, we can also observe the importance of jointly optimizing all the leaves: the sharp decreases in the red curve correspond to the optimization steps over all the leaf parameters. Since leaves account for half of the number of parameters, these significant drops in error should not be surprising (it would also be desirable to optimize multiple stump splits jointly, but since their search space grows exponentially with the number of stumps, this would make it computationally infeasible).

Despite being a simple model with relatively few parameters, an optimized stump forest tends to overfit the training data and exhibit poor generalization performance. The mid-dle plot of fig. 1 illustrates that as the number of stumps increases, the test error also rises. Initially, this is not surprising, as a higher number of stumps results in a more flexible model, which can naturally lead to overfitting. However, gradient boosting with a small learning rate (i.e., a shrinkage parameter) can produce stump forests that are much larger, yet generalize much better. In the middle plot of fig. 1, for example, the GB with 500 stumps have considerably better test error than any of the optimized stump forests, whose best test error is at around 200 stumps.

To understand better the overfitting problem, we use a simple 1D regression task where the resulting models can be easily visualized. The ground truth function is a $\sin(2\pi x)$. We generate 100 training points with added Gaussian noise to the targets. An optimized 100 stump forest achieves nearly 0 train error, and is visualized in the leftmost plot in fig. 2. Clearly, the learned model exhibits poor generalization, as it overfits the training data by interpolating all the noise, resulting in a very wiggly and discontinuous function. This can be partly attributed to the model's high flexibility: with a sufficiently large number of stumps, the training set can be perfectly fit. Additionally, the model can operate very locally, where a few stumps can significantly

alter the output in a localized region of the input while leaving the rest unchanged. Given the model's high flexibility and locality, such overfitting could be expected. (GB without shrinkage also suffers from overfitting, but with a small learning rate it does not. See the right two plots in fig. 2. This simple shrinkage method is very fundamental for the success of GB, and the theoretical understanding of it still remains unclear.)

Regularization is a fundamental technique to control overfitting. Since we know that stump forests define an additive model whose underlying geometry is a shape function per each feature, we can include regularization terms based on these shape functions. Such regularization will involve more global view of the model, thus can help in addressing the high flexibility and locality of stump forests. We define a roughness or discontinuity penalty function $r(\{\tau_t, \mu_t^l, \mu_t^r\}_{t: \phi_t=d})$: $\mathcal{P}_T(\mathbb{R}^3) \to \mathbb{R}_{\geq 0}$ that takes stumps grouped on the same feature $d$ (specifically, their 3 parameters: threshold $\tau$, left and right leaf predictions $\mu^l$ and $\mu^r$), and outputs some measure of discontinuity of the shape function $f(d)$ resulting from the stumps. Its domain is the set of triplets of size at most $T$, as there are at most $T$ stumps: $\mathcal{P}_T(\mathbb{R}^3) = \{S \subseteq \mathbb{R}^3 \mid |S| \leq T\}$. As a roughness penalty $r(\cdot)$ we use an $\ell_1$ sum of discontinuities. To express it, let us first derive the explicit form of the shape function $f_d(x_d)$ from the set of stumps. Assume, without loss of generality, that the parameters $\{\tau_t, \mu_t^l, \mu_t^r\}_{t=1}^{T_d}$ are of the stumps that use feature $d$, and that their threshold values are sorted $\tau_1 < \cdots < \tau_{T_d}$. Then the leftmost constant piece of the resulting shape function $f_d(x_d)$ is $\beta_0 = \sum_{t=1}^{T_d} \mu_t^l$, which corresponds to the left leaf predictions of all stumps (because for $x_d \in (-\infty, \tau)$, all stumps will route $x_d$ to their left child). The next leftmost piece is then $\beta_1 = \mu_1^r + \sum_{t=2}^{T_d} \mu_t^l$. More generally, the value of the $i$'s piece from the left is defined as $\beta_i = \sum_{t=1}^{i} \mu_t^r + \sum_{t=i+1}^{T_d} \mu_t^l$. The shape function $f_d(x_d)$ overall consists of the $T_d + 1$ constant pieces with values $\beta_0, \ldots, \beta_{T_d+1}$ from left to right, and with their starting intervals being $-\infty, \tau_1, \cdots, \tau_{T_d}$ (assuming all the stumps use a different threshold $\tau$). With the shape function explicitly defined, we can now express the following roughness penalty:

$$r(\{\tau_t, \mu_t^l, \mu_t^r\}_{t=1}^{T_d}) = \sum_{t=0}^{T_d-1} |\beta_{t+1} - \beta_t|. \qquad (4)$$

This penalizes the $\ell_1$ difference between the two consecutive constant pieces of the shape function $f_d(x_d)$. It is exactly zero when the shape function $f_d(x_d)$ is constant, and is a large value if $f_d(x_d)$ is very wiggly and discontinuous. It can be viewed as a first order discrete derivative of the shape function $f_d(x_d)$. Another possibility is to penalize the $\ell_2$ measure of discontinuity, but we experimentally observe the $\ell_1$ difference producing better results. A similar type of smoothness penalty occurs in fused lasso (Tibshirani et al., 2005) where the difference between neighboring coefficients is penalized, and in $\ell_1$ trend filtering for nonparametric regression (Kim et al., 2009).

Another regularization term that we propose is to penalize the deviation from the mean (or bias) for each of the leaf predictions: $\sum_{t=1}^{T} (\mu_t^l - \mu)^2 + (\mu_t^r - \mu)^2$. This encourages all stumps to be equally important, and can prevent stumps from being very dominant locally. This is conceptually similar to the idea of regularizing the leaf values proposed in (Johnson & Zhang, 2013), and currently implemented in popular gradient boosting decision tree libraries such as XGBoost and LightGBM. But instead of penalizing the actual leaf values we want to penalize their deviation from the bias. GB usually fits trees to the zero mean targets (by setting the initial constant prediction to the mean of the target), and so this type of similar regularization could already be happening in XGBoost.

With these two types of regularization, our final objective function is:

$$\min_{\Theta} \frac{1}{2} \sum_{n=1}^{N} [y_n - F(\mathbf{x}_n; \Theta)]^2 + \lambda \sum_{d=1}^{D} r(\{\tau_t, \mu_t^l, \mu_t^r\}_{t: \phi_t=d})$$
$$+ \alpha \sum_{t=1}^{T} \left((\mu_t^l - \mu)^2 + (\mu_t^r - \mu)^2\right) \qquad (5)$$

The hyperparameter $\lambda$ controls the strength of the roughness/discontinuity penalty, and we experimentally find that it is the more important one. The hyperparameter $\alpha$ controls the deviation from the bias, and plays less important role than $\lambda$. We do not tune it, and use a fixed value of $\alpha = 0.1$.

Now, for this regularized, more involved objective function, the alternating optimization steps from section 3 are still applicable, but with some changes:

**Individual stumps.** This is still solved exactly through enumeration over each (feature, threshold) pair but now the optimal values of the two leaves changes. The split finding algorithm will take into account the regularization terms when evaluating each possible split. For each potential split, the optimization problem over $\mu_l$ and $\mu_r$ is (for simplicity we show when $\alpha = 0$): $\min_{\mu_l, \mu_r} \frac{1}{2} \sum_{n \in l}(y_n - \mu_l)^2 + \frac{1}{2} \sum_{n \in r}(y_n - \mu_r)^2 + \lambda |\mu_l - \mu_r|$. Let $\bar{y}_l$ and $\bar{y}_r$ denote the sample means of the left and right partitions, $\delta = \bar{y}_l - \bar{y}_r$, and let $n_l$ and $n_r$ be the number of points in each. The optimal values of $\mu_l, \mu_r$ are:

$$\begin{cases} \mu_l = \bar{y}_l - \frac{\lambda}{n_l}, \ \mu_r = \bar{y}_r + \frac{\lambda}{n_r} & \text{if } \delta > \lambda(\frac{1}{n_l} + \frac{1}{n_l}) \\ \mu_l = \bar{y}_l + \frac{\lambda}{n_l}, \ \mu_r = \bar{y}_r - \frac{\lambda}{n_r} & \text{if } \delta < -\lambda(\frac{1}{n_l} + \frac{1}{n_l}) \\ \mu_l = \mu_r = \frac{n_l \bar{y}_l + n_r \bar{y}_r}{n_l + n_r} & \text{otherwise.} \end{cases}$$
$$(6)$$

**All leaf parameters.** Now two regularization terms will be added to the problem (4):

$$\min_{\mu, \{\mu_t^l, \mu_t^r\}_{t=1}^T} \frac{1}{2} \sum_{n=1}^N \left[ y_n - \sum_{t=1}^T \begin{array}{l} \mu_t^l \, I(x_{n\phi_t} < \tau_t) \\ + \mu_t^r \, I(x_{n\phi_t} \geq \tau_t) \end{array} \right]^2$$
$$+ \lambda \sum_{d=1}^D r(\{\tau_t, \mu_t^l, \mu_t^r\}_{t: \, \phi_t = d})$$
$$+ \alpha \sum_{t=1}^T \left( (\mu_t^l - \mu)^2 + (\mu_t^r - \mu)^2 \right) \quad (7)$$

The optimization variables $\mu, \{\mu_t^l, \mu_t^r\}_{t=1}^T$ appear either linearly or quadratically in the regularization, and the whole problem still remains convex. Instead of developing a specialized algorithm for this problem, we use a generic convex solver. In our experiments, we model the problem using this high-level formulation, and rely on CVXPY (Diamond & Boyd, 2016) to translate it into a form acceptable to a solver. In our experiments we use the MOSEK solver inside CVXPY. To accelerate training, instead of performing the optimization over the stump leaves, we can do it over the actual constant pieces of GAMs. That is, we switch from the stump forest representation to its equivalent GAM form. This change reduces the number of optimization variables by a factor of two. After solving for the constant pieces of the GAM, we convert the result back to the stump forest representation. The leftmost stump has leaf values $\mu_0^l = \beta_0$ and $\mu_0^r = \beta_1$. For subsequent stumps with index $i > 0$ we set the left leaf to $\mu_i^l = 0$ and the right leaf to $\mu_i^r = \beta_i - \beta_{i-1}$. In other words, the left leaf is always 0, and the right leaf captures the difference between adjacent constant values. It is straightforward to verify that this reconstruction yields a function equivalent to the original GAM.

The proposed regularization terms control effectively the overfitting problem. In the right plot of fig. 1 we can observe how penalizing the $\ell_1$-discontinuity helps in reducing the gap between training and test errors, and the resulting models are more accurate than the GB on the test set. In the second (from the left) plot in fig. 2 we see how these regularizations help to produce smoother stumps forests in the 1D example. We refer to our **O**ptimized **R**egularized **S**tump **F**orests as **ORSF**. We provide its pseudocode in fig. 3.

## 5. Experiments

### 5.1. Comparison on Benchmarks

Our experimental results consistently show the improved accuracy of optimized stump forests across multiple benchmarks in both classification and regression tasks. More often than not, our optimized forests require fewer number of stumps, and thus fewer number of parameters overall.

---

> **input** training set, initial forest $F(\cdot; \Theta)$ of $T$ stumps, roughness penalty $\lambda$, deviation from bias penalty $\alpha$
> **repeat**
>   Optimize all leaves jointly by solving
>   the convex problem (7)
>   **for** $t = 1$ **to** $T$
>     Optimize the individual stump $s(\cdot; \boldsymbol{\theta}_t)$
>     through enumeration
> **until** convergence
> **return** optimized forest $F(\cdot; \Theta)$

*Figure 3.* Pseudocode for Optimized Regularized Stump Forests.

We compare with the following established baselines for GAMs: explainable boosting machines (EBM) (Lou et al., 2013), gradient boosting (GB) with stumps, traditional cubic splines (implementation in Python (Servén & Brummitt, 2018)), neural additive models (NAM) (Agarwal et al., 2021), fused lasso additive models (FLAM) (Petersen et al., 2016), and FastSparse that learns piecewise constants GAMs with $\ell_0$-penalty on specially binarized features. We tune the important hyperparameters of all the baselines on a holdout set, and with the best found hyperparameters we repeat the experiment 5 times on different training/test splits to report mean and standard deviation. In our method, ORSF, for regression problems we use the squared error, and for binary classification we use the cross entropy loss with the logistic link function. The specific details of the implementation, code, hyperparameter values and the datasets are provided in appendix C.

Table 1 shows the results on regression datasets. Across all the problems, regularized stump forests achieve competitive accuracy as some of the established state-of-the-art baselines for GAMs. Importantly, among the tree/stump based methods, our approach uses far fewer number of parameters than the GB, and especially, EBM. Boosting methods work by greedily adding trees/stumps with a small learning rate, typically requiring many boosting iterations to converge. This results in a large number of trees/stumps and corresponding GAM shape functions with many constant pieces. In contrast, our approach properly optimizes a stump forest and controls its complexity with regularization. As a result, hundreds of well-optimized stumps in our method match the performance of the thousands of stumps in GB and the tens of thousands of trees in EBM.

Table 2 presents results on classification problems. The general trend is qualitatively similar to the regression case. Interestingly, traditional cubic splines show much better results here, often outperforming GB and EBM results. Splines have a rich history in statistics and possess many good theoretical properties. They also use much fewer number of parameters (degrees of freedom) while still being accurate. In the literature of GAMs, the importance of splines cannot be neglected.

*Table 1.* Train and test RMSE, model size (number of parameters) and training time (average $\pm$ standard deviation over 5 runs) for different GAMs. $N$ refers to the dataset size, $D$ is the feature dimension. Green color is the best test error, and blue is the second best.

| Dataset | | ORSF | GB | EBM | Splines | NAM | FLAM | FastSparse |
|---|---|---|---|---|---|---|---|---|
| **Cpuact** | train | 2.12±0.01 | 2.20±0.04 | 2.19±0.02 | 2.53±0.02 | 3.38±0.26 | 2.88±0.01 | 2.76±0.03 |
| $N$=8.2$k$ | test | 2.37±0.03 | 2.43±0.06 | 2.50±0.05 | 2.69±0.06 | 3.41±0.28 | 2.99±0.05 | 2.91±0.17 |
| $D$=21 | size | 642±0 | 3.4k±133 | 16.6k±36 | 271±3 | 134k±0 | 77.9k±123 | 119±4 |
| | time (s) | 9.4±0.3 | 46±17 | 39±2 | 37±0.03 | 99±1 | 85±2 | 3.8±0.5 |
| **Wine** | train ×10⁻² | 65.70±0.15 | 68.13±0.27 | 66.73±0.27 | 67.99±0.29 | 74.40±0.16 | 67.39±0.21 | 68.01±0.25 |
| $N$=6.5$k$ | test ×10⁻² | 70.02±0.66 | 70.92±0.51 | 70.12±0.39 | 71.79±1.40 | 76.07±2.11 | 70.19±0.84 | 71.77±0.63 |
| $D$=11 | size | 724±12 | 770±32 | 3.9k±11 | 197±7 | 70.1k±0 | 5041±11 | 182±4 |
| | time (s) | 6.0±0.3 | 2.87±0.58 | 4.44±1.33 | 56±16 | 64±0 | 53±3 | 0.57±0.07 |
| **Housing** | train ×10⁻² | 51.84±0.16 | 54.24±0.27 | 52.70±0.04 | 53.37±0.21 | 71.56±0.30 | 55.08±0.20 | 54.62±0.20 |
| $N$=21$k$ | test ×10⁻² | 54.80±0.65 | 56.15±0.58 | 55.23±0.68 | 55.49±0.61 | 72.23±0.88 | 56.24±0.74 | 56.29±0.65 |
| $D$=8 | size | 1.4k±20 | 2.4k±31 | 7.2k±8 | 528±2 | 51.0k±0 | 118k±101 | 579±9 |
| | time (s) | 13.6±0.4 | 42±8 | 36±2 | 37±2 | 175±2 | 73±2 | 3.94±0.73 |
| **Diamond** | train ×10² | 9.95±0.02 | 10.07±0.05 | 10.11±0.03 | 10.02±0.02 | 13.53±0.22 | 11.75±0.03 | 10.01±0.02 |
| $N$=54$k$ | test ×10² | 10.15±0.08 | 10.19±0.08 | 10.23±0.06 | 10.96±1.45 | 13.59±0.25 | 11.70±0.12 | 10.17±0.09 |
| $D$=26 | size | 934±16 | 1182±81 | 3.4k±7 | 273±24 | 86k±0 | 4139±12 | 516±11 |
| | time (s) | 25.1±0.9 | 140±58 | 20±2 | 42±0.4 | 708±2 | 805±11 | 45±10 |
| **Year** | train | 9.12±0.03 | 9.30±0.03 | 7.53±0.02 | 9.14±0.03 | 10.22±0.05 | | 9.14±0.03 |
| $N$=423$k$ | test | 9.30±0.01 | 9.35±0.00 | 9.82±0.02 | 9.38±0.03 | 10.22±0.08 | out of time | 9.29±0.01 |
| $D$=90 | size | 1379±0.8 | 1490±25 | 368k±0 | 2158±55 | 573k±0 | > 2 days | 2601±63 |
| | time (s) | 1402±43 | 4368±256 | 4262±437 | 3618±45 | 8858±88 | | 973±65 |
| **FPS** | train | 55.40±0.09 | 55.48±0.09 | 55.42±0.09 | 55.41±0.09 | 56.23±0.10 | | 55.41±0.09 |
| $N$=401$k$ | test | 55.41±0.34 | 55.45±0.34 | 55.42±0.34 | 55.42±0.34 | 55.62±0.24 | out of time | 55.42±0.34 |
| $D$=100 | size | 983±37 | 824±57 | 2372±12 | 411±1 | 288k±0 | > 2 days | 1250±17 |
| | time (s) | 798±21 | 1803±466 | 655±84 | 2043±2 | 4397±10 | | 625±10 |

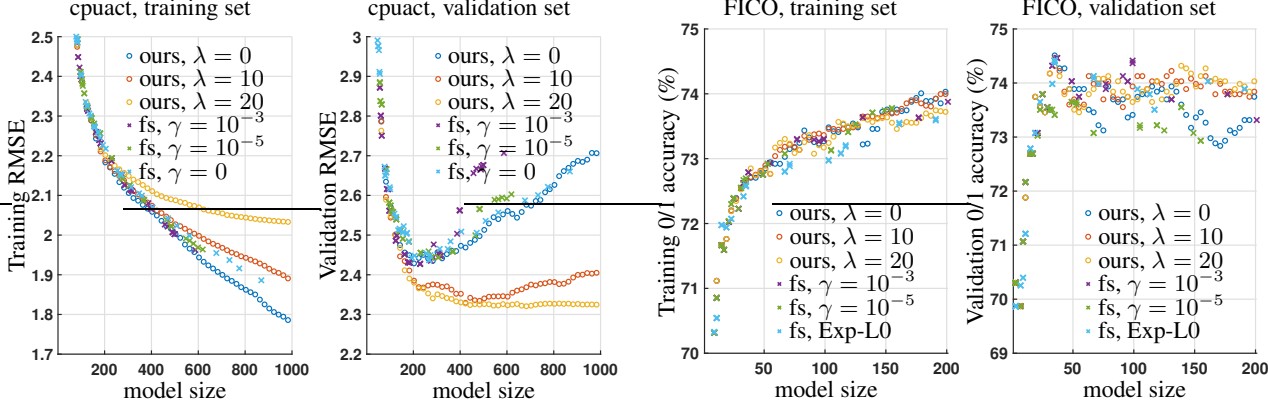

*Figure 4.* Comparison on the CPU Activity regression and FICO classification datasets of the regularization paths generated by our method versus FastSparse (fs).

### 5.2. Regularization Paths over Number of Stumps

In our approach, we construct a "regularization" path by starting with a small number of stumps and incrementally adding more, re-optimizing the entire model at each step. In fig. 4, we compare the regularization paths of our method with those of FastSparse, which generates its path by varying a sequence of $\ell_0$-penalty parameters. On the cpuact regression task, our method achieves superior generalization performance, benefiting from the $\ell_1$-roughness regularization penalty. For the FICO dataset, the performance of both methods is comparable. It is worth noting that although both methods generate 100 models along the regularization path, FastSparse typically produces fewer unique models,

as multiple $\ell_0$-penalty values often result in the same solution. In contrast, our method provides more direct control over model size through the number of stumps $T$, effectively enforcing an $\ell_0$ constraint. Our regularization path includes 100 distinct models, though only 50 are displayed in the figure to maintain clarity. More detailed results of this comparison is available in Appendix fig. 7 and fig. 8.

### 5.3. Effect of Different Regularizations

In fig. 5, we examine the individual effects of three different types of regularizations on stump forests using the CPU activity dataset. As expected, all regularization methods exhibit typical behavior: increasing the regularization

Table 2. As in Table 1 but for classification datasets. The error is a 0/1 misclassification (%).

| Dataset | | ORSF | GB | EBM | Splines | NAM | FLAM | FastSparse |
|---|---|---|---|---|---|---|---|---|
| **Letter** | train | 15.94±0.14 | 16.38±0.17 | 16.12±0.20 | 15.87±0.14 | 21.54±1.1 | 17.94±0.18 | 15.88±0.14 |
| $N$=20$k$ | test | 16.40±0.52 | 16.88±0.41 | 16.63±0.42 | 16.55±0.70 | 22.53±1.88 | 17.95±0.51 | 16.57±0.67 |
| $D$=16 | size | 403±13 | 420±15 | 502±2 | 224±1 | 68k±0 | 510±2 | 399±5 |
| | time (s) | 14.9±0.2 | 32±3 | 31±1 | 58±2 | 153±0 | 71±1 | 18±2 |
| **Churn** | train | 18.88±0.19 | 19.00±0.23 | 18.84±0.08 | 18.78±0.15 | 22.59±2.13 | 19.85±0.18 | 18.88±0.11 |
| $N$=7.0$k$ | test | 19.28±0.29 | 19.32±0.37 | 19.47±0.51 | 19.32±0.48 | 21.69±2.02 | 20.30±0.88 | 19.87±0.36 |
| $D$=45 | size | 129±5 | 644±48 | 7292±11 | 40±0.04 | 120k±0 | 13.7k±15 | 105±8 |
| | time (s) | 6.8±0.4 | 3±1 | 15±1 | 0.5±0.03 | 120±2 | 113±2 | 0.59±0.07 |
| **FICO** | train | 24.86±0.13 | 26.54±0.15 | 26.37±0.10 | 26.79±0.15 | 28.23±0.41 | 27.15±0.21 | 25.87±0.16 |
| $N$=10$k$ | test | 27.33±0.04 | 27.62±0.30 | 27.43±0.31 | 27.35±0.17 | 28.08±0.61 | 27.64±0.52 | 27.80±0.33 |
| $D$=23 | size | 550±28 | 1002±66 | 3680±9 | 83±1 | 130k±0 | 3791±11 | 196±10 |
| | time (s) | 7.8±0.1 | 1.6±0.6 | 7±0.2 | 1.96±0.10 | 180±1 | 61±1 | 1.74±0.12 |
| **IJCNN** | train | 4.42±0.05 | 4.56±0.07 | 4.51±0.03 | 4.44±0.04 | 7.51±0.44 | 6.86±0.08 | 4.84±0.16 |
| $N$=50$k$ | test | 4.95±0.14 | 5.10±0.15 | 5.00±0.14 | 4.92±0.20 | 7.48±0.55 | 7.14±0.15 | 5.52±0.21 |
| $D$=22 | size | 414±23 | 918±21 | 12.3k±0 | 266±0.5 | 101k±0 | 828k±242 | 883±18 |
| | time (s) | 46±1 | 148±24 | 19±0 | 153±40 | 501±1 | 249±6 | 47±1 |
| **Covtype** | train | 22.50±0.03 | 22.56±0.02 | 22.46±0.02 | 22.48±0.02 | 26.16±0.50 | | 22.49±0.02 |
| $N$=581$k$ | test | 22.71±0.11 | 22.77±0.10 | 22.68±0.12 | 22.72±0.10 | 26.08±0.54 | out of time | 22.68±0.10 |
| $D$=54 | size | 504±4 | 1090±32 | 6402±4 | 403±1 | 170k±0 | > 2 days | 841±15 |
| | time (s) | 1091±16 | 1202±49 | 325±5 | 15624±84 | 5373±16 | | 2763±177 |
| **Bank** | train | 9.81±0.04 | 10.00±0.03 | 9.75±0.05 | 9.79±0.04 | 10.09±0.08 | 11.27±0.04 | 9.79±0.04 |
| $N$=41$k$ | test | 9.83±0.17 | 9.99±0.13 | 9.91±0.17 | 9.88±0.12 | 9.87±0.26 | 11.23±0.15 | 9.86±0.14 |
| $D$=62 | size | 231±4 | 530±15 | 1103±7 | 95±2 | 174k±0 | 1182±1 | 64±4 |
| | time (s) | 47±2 | 34±7 | 40±3 | 22±2 | 662±10 | 916±3 | 19.6±3.3 |

Figure 5. The effect of different regularization types in stump forests for the cpuact dataset. The number of stumps $T = 200$. For our final method we use the combination of $\ell_1$-discontinuity and the deviation from the bias regularizations.

penalty initially results in high test error due to overfitting, followed by a decrease to the lowest test error indicating the best generalizable models, and finally, an increase in test error due to underfitting as the penalty becomes too strong. We can also observe the good ranges of the hyperparameter values, for example, the deviation from the bias penalty typically favors small values of the penalty parameter $\alpha$. When comparing the importance of these 3 methods on the test error, the $\ell_1$-roughness penalty usually produces models with the lowest test error than the other two. In our main experiments we use the combination $\ell_1$-discontinuity and the deviation from the bias regularizations.

## 5.4. Case Study: Car Price Prediction

To showcase the interpretability of stump forest additive models, we use the problem of predicting the price of a used car. From the Kaggle platform we obtain 100,000 UK

Used Car Data set, which features used car listings in UK, grouped by different car manufactures. To avoid the large number of car make/model categories when visualizing, we focus on the following 6 popular models: BMW 3 series, Mercedes C Class, Volkswagen Golf, Ford Focus, Hyundai Tucson and Toyota Yaris. The total number of training points is 15,200 with additional 3,800 instances used for testing. The following 8 features describe a given car: the year of manufacture, mileage (a total number of miles traveled), amount of yearly tax, miles per gallon (MPG), engine size, a make/model, a transmission type and a fuel type. The first 5 features are numerical, and the last 3 are categorical.

We train our forest with 700 stumps and a roughness penalty of $\lambda = 60$. The model achieves a mean absolute error (MAE) of £1,600 on the test set. Considering the av-

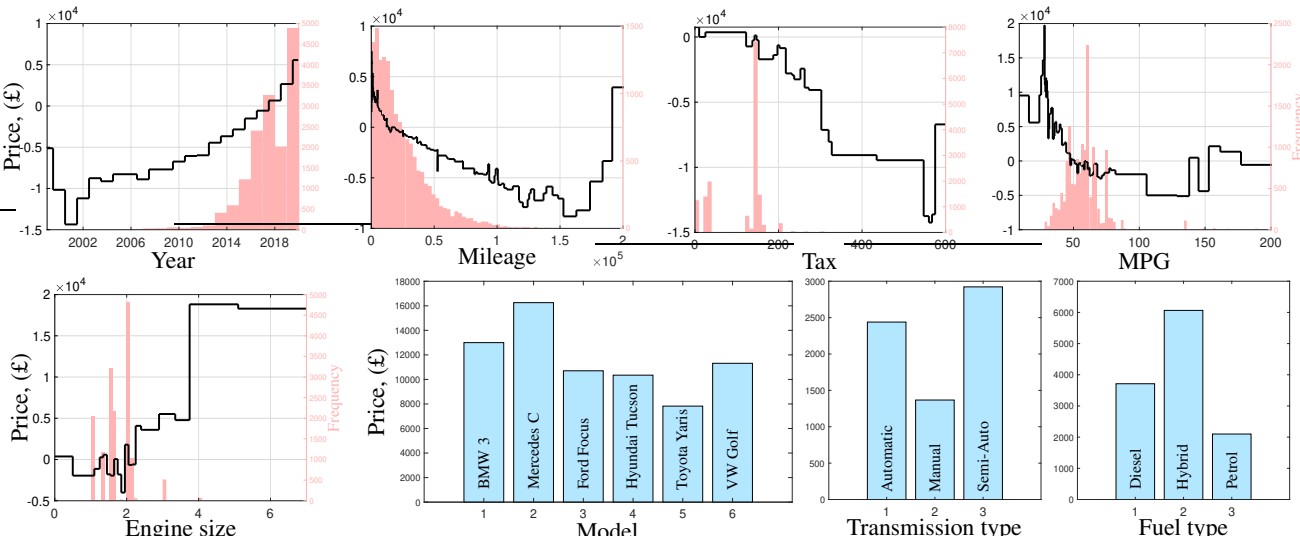

*Figure 6.* Visualization of the resulting additive model shape functions from our optimized stump forests for the UK used car dataset. For the numerical features, the light red bars show the histogram of the training points with the frequency values given on the right $y$-axis.

erage price is £16,900 with a standard deviation of £7,600 on the whole dataset, this error margin is relatively acceptable. And the model is much more accurate than a linear regression which has a test MAE of £2,200. Most importantly, we can visualize the entire model as 8 plots for the 8 used features in fig. 6, and interpret it. There was also a constant bias term, but we eliminate it by distributing its value to the 3 categorical features.

We expect newer cars to be more expensive, and the shape function of the year feature generally confirms this expectation. There is an unexpected negative correlation with prices for the years 1999-2001, but this anomaly is due to having only six car listings from those years. Another crucial feature in predicting car prices is mileage; typically, cars with higher mileage are cheaper. The shape function for mileage supports this relationship. However, there is an unusual slight drop in price around 50,000 miles. A closer examination of the dataset reveals an outlier: a Mercedes C-Class with 52,700 miles listed at £2,140, whereas the normal price for this model is about £10,000. Miles per gallon (MPG), which characterizes fuel consumption, is another significant predictor variable. Its shape function generally shows a negative relationship with price. In the training set, we find newer Mercedes C-Class cars with engine sizes of 4.0 and 6.3 that have low MPG but are very expensive. Additionally, there are noisy points with mislabeled MPGs, such as eight newer BMW 3 Series cars with an MPG of 8.8, which should have been 30.0. Regarding categorical features, the distribution of prices within each category appears to be reasonable: luxury brands like Mercedes and BMW are more expensive, cars with manual transmissions tend to be cheaper, and hybrid fuel types are usually more expensive. Overall, by visualizing and analyzing the

model, we can understand it, and possibly even debug and correct the parameters and/or the dataset. This example on car price prediction clearly demonstrates the inherent interpretability of additive models. Appendix figs. 9 and 10 show the shape functions for the other GAMs, EBM and PyGAM. Overall, the general trend in EBM shapes is quite similar to the ones obtained by our method and also being similar in accuracy (around 2,200 test RMSE), although it generated an overly noisy curve for the mileage feature. PyGAM, on the other hand, generates very smooth curves, and behaves quite unpredictably in regions with less or no data.

## 6. Conclusion

We have presented a novel approach to learning forests of axis-aligned decision stumps, with its equivalent view of a generalized additive model. Instead of bagging and boosting, we use alternating optimization to learn a fixed-size stump forest. This makes it possible to optimize exactly at each step the stump parameters by enumeration and jointly optimize the leaf values by solving a convex problem. To mitigate overfitting, we introduce effective regularization techniques that take into account the global smoothness of the model when viewed as a piecewise-constant function. These regularized stump forests exceed the accuracy of state-of-the-art GAM methods, yet with fewer parameters. Our approach is pioneering in that it learns stump forests without the need for traditional ensembling techniques such as bagging or boosting, and in that it can be initialized from an existing forest, thus making retraining possible. A possible future research direction is to define global smoothness-based regularization methods to learn a more general, parametric decision forest of a fixed size.

## Acknowledgements

Work partially supported by NSF award IIS–2007147.

## Impact Statement

This paper presents work whose goal is to advance the field of Machine Learning. There are many potential societal consequences of our work, none which we feel must be specifically highlighted here.

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

## A. Limitations

In this paper we focus only on univariate shape functions $f_d(x_d)$, but using bivariate interactions $f_{ij}(x_i, x_j)$ could provide more accurate results. However, such pairwise terms are somewhat less interpretable, and the total number of such terms grows quadratically with the feature dimension. Perhaps extending this work to include depth-2 trees that model bivariate interactions can be one possible future direction. One possible limitation is in comparison with traditional splines that typically use fewer number of parameters. However, tree/stump-based methods tend to be more accurate as demonstrated here, and in the previous literature (Lou et al., 2012). And compared with models of the same family of stumps/trees (GB and EBM), our models are much smaller.

## B. Additional Experiment Results

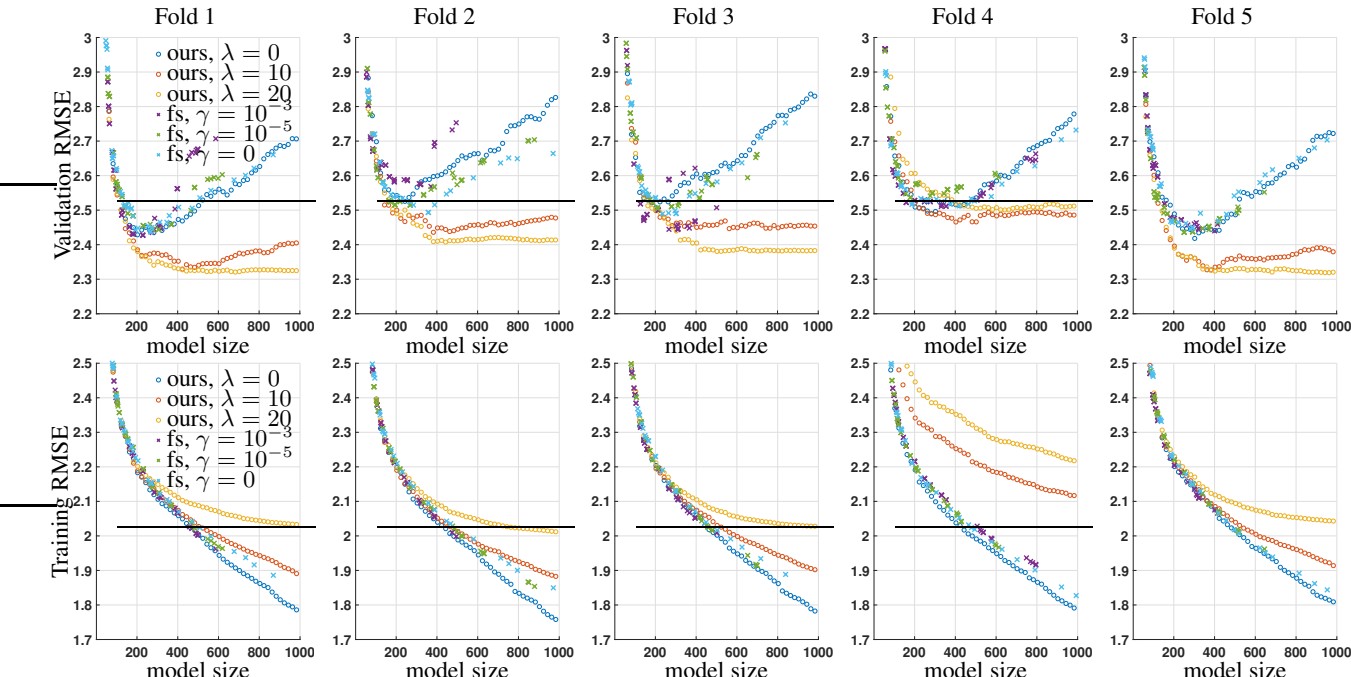

*Figure 7.* Five-fold cross-validation comparison on the CPU Activity regression dataset of the regularization paths generated by our method versus FastSparse (fs). In our approach, the regularization path is constructed by starting with an empty forest and incrementally adding 5 stumps at a time, re-optimizing the entire forest at each step using our algorithm. We present three curves corresponding to different values of the roughness penalty parameter $\lambda \in \{0, 10, 20\}$. For FastSparse, we show results for three different values of the parameter $\gamma \in \{0, 10^{-5}, 10^{-3}\}$. We set the `num_lambda` parameter to 100 and `max_support_size` to 1000, which generates a regularization path across 100 different `lambda_0` values. However, the actual number of unique models produced by FastSparse is typically lower, as multiple values of `lambda_0` often lead to the same model. In contrast, our method allows for more direct control over model size via the number of stumps $T$, effectively imposing an $\ell_0$ constraint. Our regularization path consists of 100 distinct models, although only 50 are shown in the figure to avoid visual clutter. The model size here is defined as the number of thresholds times 2 (a constant piece value and a threshold) plus 1 for the bias.

## C. Experiment Details

For all algorithms, including ours, we tune the important hyperparameters on a holdout set, and with the best found hyperparameters perform 5 experiments on different train/test splits to report mean and standard deviation.

**Implementation of our algorithm**   We implement our algorithm in the combination of Python and C++. The optimization step over each stump is implemented in C++. The convex problem of joint leaf fitting is implemented in Python. We use the CVXPY open source Python package (Diamond & Boyd, 2016) to model the leaf fitting problem using high-level formulation, and rely on the library to convert/compile it into a specific solver form. As a convex solver inside CVXPY, we use MOSEK, a large scale optimization software. We use CVXPY version 1.4.3, and MOSEK version 10.1. We per-

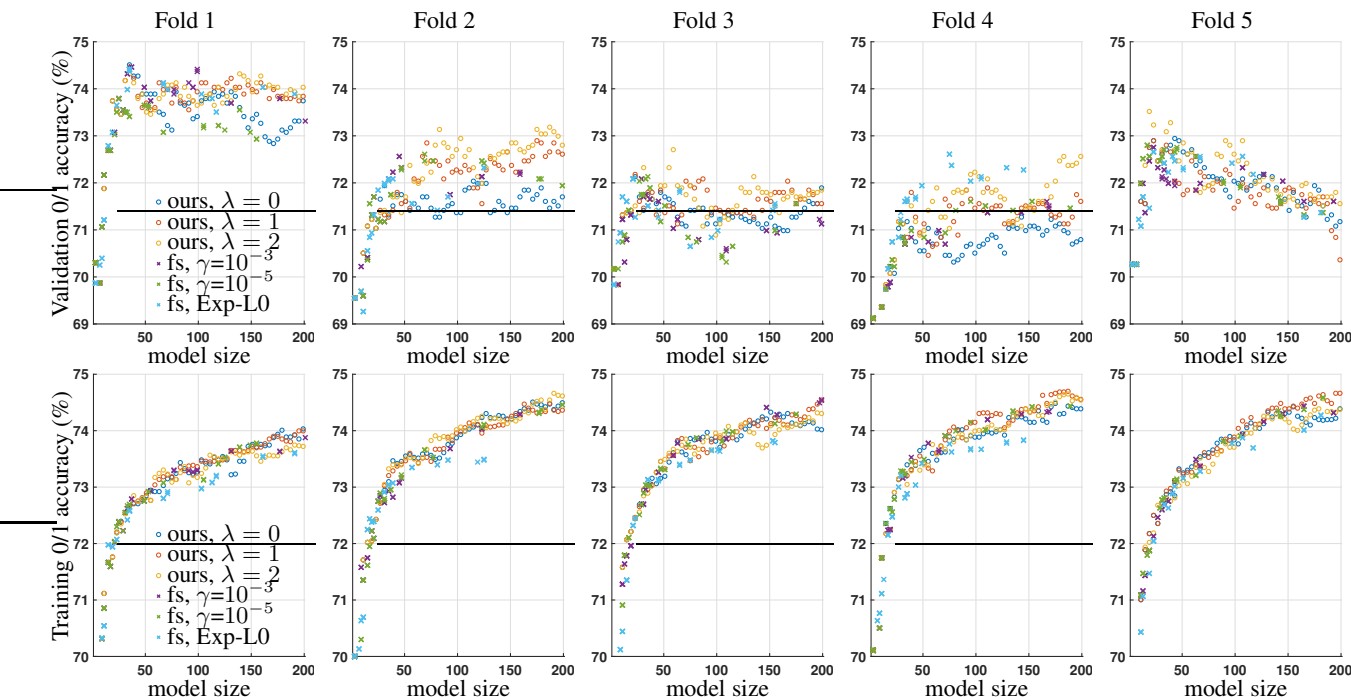

*Figure 8.* Same setup as Fig. 7, but applied to the FICO classification dataset. For our method, we generate the regularization path by incrementally adding one stump at a time and re-optimizing the current forest using our algorithm, continuing until the forest contains 100 stumps. To reduce visual clutter, we display only 50 points from our regularization path. For FastSparse (fs), following Fig.8 from their paper, we report results for logistic loss with two different values of $\gamma \in \{10^{-5}, 10^{-3}\}$, and for exponential loss with `penalty=L0`. We run the regularization path with `num_lambda=100`, and similarly as with fig. 7, we obtain a fewer number of unique models because multiple values `lambda_0` produce the same result.

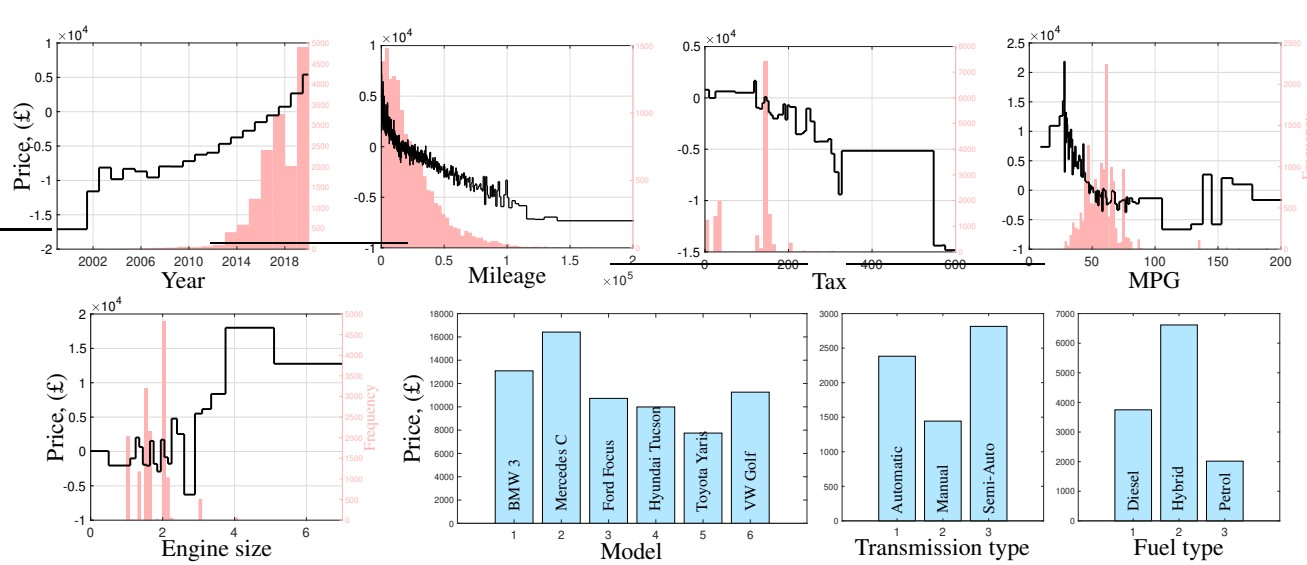

*Figure 9.* As fig. 6, but for EBM shape functions.

form grid search on the following hyperparameter values: number of stumps = $\{200, 400, 600, 800\}$, roughness penalty $\lambda = \{2.0, 4.0, 6.0\}$ for classification datasets, $\lambda = \{20.0, 40.0, 60.0\}$ for regression datasets. We do not tune the deviation from bias hyperparameter $\alpha$, and use the fixed value $\alpha = 0.1$. As an initial stump forest we use stumps with random parameters.

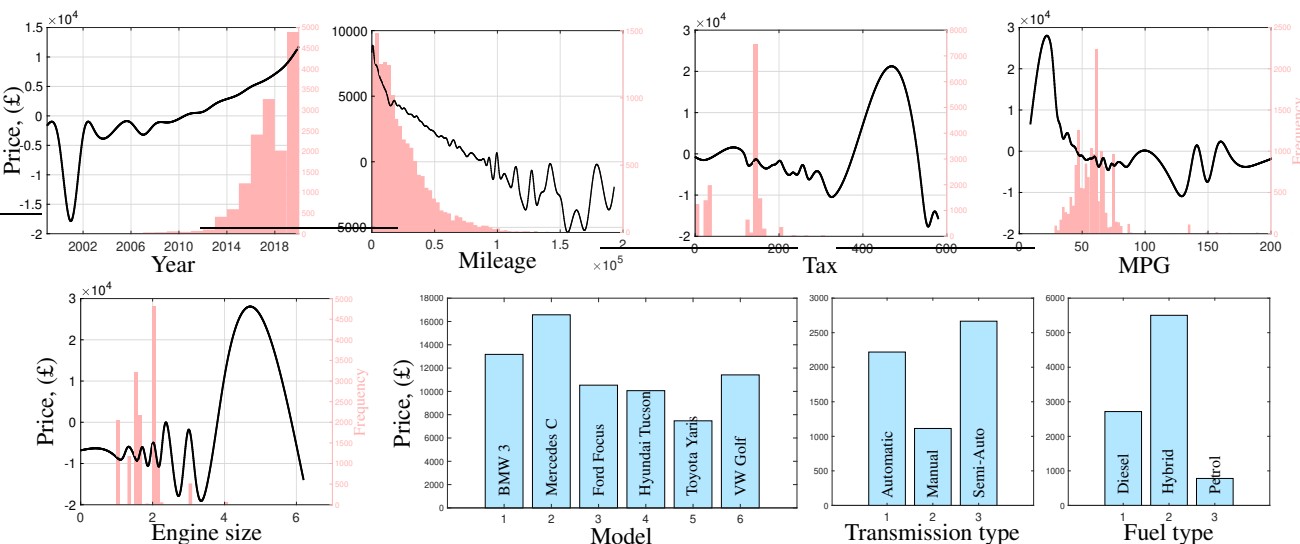

*Figure 10.* As fig. 6, but PyGAM shape functions.

**Hardware**   Except for the Neural Additive Model (which is trained on a GPU), all experiments are performed on a CPU Intel(R) Xeon(R) CPU E5-2699 v3 @ 2.30GHz, 128 GB RAM.

The specific details of the baselines and tuned hyperparameters:

**Gradient Boosting (GB):**   we use the scikit-learn's implementation: GradientBoostingRegressor and GradientBoosting-Classifier (Pedregosa et al., 2011). We set the maximum depth to 1 and perform grid search on the learning rate {0.01, 0.05, 0.1, 0.3}. We set the number of boosting iterations (`n_estimators`) to a very high number ($10^6$), and use early stopping based on a validation set with `n_iter_no_change` equal to 100. The version of the scikit-learn is 1.4.2.

**Explainable Boosting Machine (EBM):**   we use the official implementation from the `interpret` Python library[1]. We perform grid search on the following hyperparameter values: `learning_rate`: {0.005, 0.01, 0.05}, `max_bins`: {512, 1024, 2048}, `min_samples_leaf`: {2, 4, 8}. We set the `interactions` parameter to 0 to use only univariate terms. `max_rounds` is set to 25 000 with `early_stopping_rounds` 50. The version of the `interpret` package is 0.6.1.

**Splines:**   we use the PyGam package[2] in Python. We perform grid search on the following hyperparameters: strength of the smoothing penalty $\lambda = \{0.001, 0.01, 0.1, 1, 10, 100, 1000, \}$, and the number of knots = $\{10, 20, 50, 100\}$. The version of the PyGam package is 0.9.1.

**Neural Additive Model (NAM):**   we use the paper's (Agarwal et al., 2021) official implementation[3]. As in their paper, we try two types of architectures: MLP with 64-64-32 ReLU layers and 1024 Exu layer. We perform grid search on the following hyperparameters: dropout = {0.0, 0.1, 0.3, 0.5}, weight decay = $\{10^{-6}, 10^{-5}, 10^{-4}\}$, learning rate = {0.001, 0.01, 0.1}. We use a batch size of 1024 and train for 100 epochs. Only for this baseline we train it on an NVIDIA TITAN V GPU.

**Fused Lasso Additive Model (FLAM):**   is a nonparametric model that uses a discrete first derivative matrix to penalize the roughness of the piecewise constant predictions. We use the official implementation[4] in R. We set the $\alpha$ parameter to 1 as feature selection is not the goal here, and only tune the roughness penalty $\lambda$ from their default range of 50 values.

---

[1] https://github.com/interpretml/interpret
[2] https://github.com/dswah/pyGAM
[3] https://github.com/google-research/google-research/tree/master/neural_additive_models
[4] https://cran.r-project.org/web/packages/flam/index.html

**FastSparse:** We use the Python package `fastsparsegams`. We use the method `convert_continuous_df_to_binary_df` to binarize the features. We perform grid-search on the following values of hyperparameters: `algorithm = [CD, CDPISI]`, `max_support_size = [10, 100, 1000, 10000]`, and the 100 $\lambda$ values in regularization path (we set `num_lambda = 100`, and scan through the `model.lambda_0` values after training). For the `penalty` parameter, we use `L0`. For classification, we use `Logistic` as the `loss`,

**Estimation of the number of parameters.** For piecewise-constant models (ours, GB and EBM), we estimated the number of parameters as the number of constant pieces times 2 summed over each shape function. This is because we must store the piece value, and where the piece ends. For the neural networks (NAM), we just report the number of parameters in the network architecture. And for the FLAM, because it is non-parametric, we report the size of the training set, more specifically, the number of unique feature values summed over per each dimension.

**Regression dataset description:**

**Cpuact** The goal is to estimate the percentage of time a CPU operates in user mode. The features are various statistics related to memory and other activities. Source is the Delve project: http://www.cs.toronto.edu/~delve/data/comp-activ/desc.html.

**Wine** The task is to predict the wine quality from 0 (very bad) to 10 (very excellent) based on the attributes as acidity, sugar, chlorides, etc. Obtained from the UCI ML repository (Lichman, 2013).

**Housing** California Housing dataset, a standard regression benchmark. Obtained through the scikit-learn's `fetch_california_housing` function.

**Diamond** The task is to predict the price of a diamond based on the characteristics of clarity, carat weight, etc. We download it from the OpenML dataset repository: https://www.openml.org/search?type=data&sort=runs&id=42225&status=active.

**Year** predicting the release year of a song from audio features. Obtained from the UCI ML Repository (Lichman, 2013).

**FPS** the goal is to predict the frames-per-second in video games based on characteristics of a CPU, GPU and the game itself. Sourced from the OpenML repository: https://www.openml.org/search?type=data&status=active&id=42737. We only use the "userbenchmark" source, and discard the "fpsbenchmark" source. We also drop the columns with missing values. We perform one-hot-encoding for the categorical variables.

**Classification dataset description:**

**Letter** English letter recognition task, available in the UCI ML Repository (Lichman, 2013). We convert it into binary classification by combining the letters A-M into one class and the rest to the other class.

**Churn** The binary classification task to predict which customer will churn. The features are various characteristics of the customer. Obtained from Kaggle: https://www.kaggle.com/datasets/blastchar/telco-customer-churn.

**FICO** Dataset from the Explainable Machine Learning Challenge. The task is to predict whether a consumer pays the credit on time or is overdue. The features come from an anonymized credit bureau data. Obtained from the official website: https://community.fico.com/s/explainable-machine-learning-challenge.

**IJCNN** The dataset comes from an IJCNN 2001 competition. We obtain it from the LIBSVM binary dataset collection: https://www.csie.ntu.edu.tw/~cjlin/libsvmtools/datasets/binary.html.

**Covtype** Classification task of pixels into 7 forest cover types based on attributes such as elevation, aspect, soil-type, slope, etc. We turn it into binary classification by as the majority class vs the rest. Sourced from the UCI ML Repository (Lichman, 2013).

**Bank** the task to predict whether a marketing campaign of a banking institution is successful or not. The features are various characteristics of a customer. Obtained from the UCI ML Repository (Lichman, 2013).

