# OpenReview forum: "Generalized additive models via direct optimization of regularized decision stump forests"
_ICML.cc/2025/Conference — ICML 2025 poster_

### Official Review · Reviewer_owkV · 2025-03-09

**Overall Recommendation:** 4

**Summary:**

The authors propose an alternative framework to train GAM models with piecewise constant shape functions based on an alternating optimisation algorithm. A specific regularization strategy is proposed to deal with the issue of overfitting when directly optimizing this type of models.
The authors find that this approach is competitive with other boosting based approaches to optimize the same type of models.

**Claims And Evidence:**

As far as I am concerned, the authors make no unsupported claims.

I think, however, that the authors could clarify that even though their method converges, assuming no cycles, there is no guarantee of convergence to the global optimum solution. Furthermore, there is no guarantee on the rate of this convergence. This type of method could converge very slowly in problems with narrow "diagonal" valleys where alternating minimization makes little progress with each iteration.

**Essential References Not Discussed:**

None that I am aware.

**Experimental Designs Or Analyses:**

The choice of baselines and the experimental setup all seem adequate.

**Methods And Evaluation Criteria:**

In my opinion, the methods and evaluation make sense for the problem at hand.

**Other Comments Or Suggestions:**

None

**Other Strengths And Weaknesses:**

### Strengths

The authors put a great effort in the presentation of the paper. Everything is clearly explained, with plenty of insightful figures that illustrate the main points.

Furthermore, the proposed approach, while somewhat reminiscent of backfitting, is interesting and original, presenting a significant enough departure from how similar piece-wise constant models are typically trained.

### Weaknesses

My overall interpretation of the results is that GAM models seem rather limited in their representation capacity given their rigid model structure, and there isn't much to be gained in the choice of different families of shape functions or how they are fit.
Particularly, for all of the models learning piecewise constant shape functions, the differences in performance seem small, specially when comparing with black-box models that don't follow the GAM structure.
As an example, a standard GBDT (with deeper trees) achieves < 3% error on Covtype. This makes the small differences between GAM models, all of which achieve ~22% error, seem inconsequential in practice.

As such, while I value work on interpretable models, I don't see this work as having a particularly large impact, specially given that it seems significantly slower to fit than alternatives.

**Post rebuttal note:** Considering the authors' efforts in optimizing their algorithm and the significant speedups they achieved, this pointed weakness is no longer applicable and the argument for adopting the proposed algorithm becomes considerably more compelling.

**Questions For Authors:**

(**RESOLVED**) 1. As the authors point out as a possible point of improvement, it seems to me that the optimization of the leaf values could be formulated as a standard quadratic program. Perhaps calling a solver directly would forego the overhead of CVXPY and speed up the training process which is one the weaker points of the algorithm. If a substantial speed up could be achieved I would be willing to raise my score to a clear accept.

**Relation To Broader Scientific Literature:**

GAM models that rely on piecewise constant shape functions, such as Explainable Boosting Machines, are arguably SoTA within this model class. This work proposes a new approach to train models using the same type of shape function, through directly optimizing the splits of a stump forest rather than the boosting approach of EBM.

**Theoretical Claims:**

There are no proofs or theoretical claims.

---

> ### Author Rebuttal · Authors · 2025-04-01
>
> Thank you for providing us with a very valuable and insightful review! We greatly appreciate your effort in reviewing our paper.
>
> * **Convergence to global optimum.** The problem of learning an optimal decision stump forest is computationally intractable in general. Even in the case of 1D splines, finding optimal knot positions is known to be NP-hard [1]. Therefore, guaranteeing convergence to a global optimum in polynomial time is unlikely. Our algorithm guarantees a monotonic decrease in the objective function at each iteration and converges to a local optimum. We expand on this discussion in our rebuttal to wGmH.
>
> * **Slow rate of convergence.** Our optimization space includes both continuous and discrete components. The continuous part, the leaf values, is optimized exactly by solving a convex problem. The discrete part involves the split parameters of the stumps. Alternating optimization is one of the most effective approaches for such problems. For instance, the well-known k-means algorithm also alternates between discrete and continuous updates and is rarely hindered by slow convergence in practice. In our experiments, the algorithm typically converges within around 5 iterations, where one iteration is defined as a full pass over all stumps and an update of all leaf parameters.
>
> * **Limited representation capacity.** We acknowledge and agree with the observation regarding the limited representation capacity of GAMs, and that there may be limited room for improvement over existing baselines. While it is true that on datasets like Covertype, GAMs perform notably worse than high-capacity models, there are also cases, such as the cpuact dataset, where GAMs achieve accuracy comparable to models like XGBoost. The key strength of GAMs lies in their interpretability and their ability to provide meaningful insights into the data. This is particularly valuable in sensitive domains like healthcare, where understanding model behavior is crucial [2].
>
> * **Training time improvement.** The majority of our time during this research project was devoted to analyzing the behavior of the algorithm and exploring various strategies to address the overfitting. Comparatively little time was initially spent on improving runtime performance. During the rebuttal period, we revisited the formulation of the leaf-fitting problem and focused on optimizing its efficiency.
>
>   Instead of optimizing over the stump leaf parameters, we now directly optimize over the constant piece values of the GAM shape functions. That is, we switch from the stump forest representation to its equivalent GAM form. This change reduces the number of optimization variables by a factor of two. Additionally, by rewriting the convex problem in CVXPY using a different set of atomic functions, we were able to switch the underlying solver to OSQP (Operator Splitting Quadratic Program). In line with your suggestion, this allows us to cast the problem as a standard quadratic program while still using CVXPY to handle the transformation, with minimal overhead. These changes result in a significant improvement in training time, by a factor of 5 to 20 depending on the dataset.
>
>   After solving for the constant pieces of the GAM, we convert the result back to the stump forest representation. Specifically, let the shape function for feature $d$ be $f_d(x_d)$ which consists of the $T_d+1$ constant pieces with values $\beta_0, \dots, \beta_{T_d+1}$ from left to right. The leftmost stump has leaf values  $\mu^l_0 = \beta_0$ and $\mu^r_0 = \beta_1$. For subsequent stumps with index $i > 0$ we set the left leaf to $\mu^l_i = 0$ and the right leaf to $\mu^r_i = \beta_i - \beta_{i-1}$.  In other words, the left leaf is always 0, and the right leaf captures the difference between adjacent constant values. It is straightforward to verify that this reconstruction yields a function equivalent to the original GAM. The updated Tables 1 and 2 (https://anonymous.4open.science/r/pdf_to_anon-1CD8/icml25_rebuttal_figures.pdf or https://drive.google.com/file/d/1Xso8hwFd9rB_XqAzH6H8ri7vUbrXEaq6/view?usp=sharing) reflect the significantly improved training times achieved through this new formulation.  Now, our method is among the fastest to learning accurate piecewise constant GAMs.
>
> * **Warm starting ability.** Another independent strategy to accelerate training and potentially improve accuracy is to initialize the optimization from stronger starting points. During the rebuttal period, we incorporated an additional baseline, FastSparse, which efficiently produces reasonably accurate piecewise constant GAMs. Using these models for initialization led to even faster convergence and slightly improved accuracy on several datasets.
>
> [1] G. Beliakov, "Least squares splines with free knots: global optimization approach", in Applied Mathematics and Computation, 2004.
>
> [2] R. Caruana et al., "Intelligible Models for HealthCare: Predicting Pneumonia Risk and Hospital 30-day Readmission", KDD 2015.

---

> > ### Comment · Reviewer_owkV · 2025-04-02
> >
> > After carefully reading the remaining reviews and the authors' rebuttals, I have no further serious concerns about the paper.
> > I appreciate the authors' efforts in optimizing their algorithm and achieving significant speed-ups. I also appreciate the comparisons with the baselines requested by reviewer `r73Z`. As such, I am happy to increase my score from 3 to 4.
> >
> >
> > **Minor point:**
> > My comment about convergence to a local optimum was merely that a clarification could be added to the manuscript, in the paragraph starting on line 158, when the authors claim that the algorithm converges. While I agree that it should be obvious, given the nature of the problem, I could see a reader being mislead since the authors don't explicitly mention that this convergence is to a local optimum.

---

### Official Review · Reviewer_r73Z · 2025-03-13

**Overall Recommendation:** 3

**Summary:**

This paper proposes an alternating optimization method for training an additive model composed of decision stumps. The optimization procedure alternates between two steps: (1) selecting a feature and determining the optimal split value for a decision stump, and (2) jointly optimizing all coefficients through a convex optimization solver. The method incorporates a regularization term to penalize the roughness of the coefficients, which helps control model complexity. The authors claims that their approach effectively reduces the number of stumps while maintaining both training and test performance.

## update after rebuttal

I want to thank the authors for replying to my second round rebuttal comments. I have also read the reviews and rebuttal comments by other two reviewers.

The mathematics behind individual stump optimization makes sense to me. This should be included in the paper (at least in the Appendix) during revision.

The new plot on the FICO dataset (Figure 2) included in the anonymous link also looks reasonable to me. This figure should be included during revision. I think the reason for FastSparse of having a larger support size than I expected is on your Fold 4, somehow the sparsity wasn't optimized around 20 features. This wasn't my experience, but maybe it has something to do with your particular 5CV split or the feature binarization preprocessing step.

Since my major concerns are addressed, I no longer have a big problem against this paper getting accepted. I am raising my evaluation to score of 3, above the acceptance threshold.

**Claims And Evidence:**

The claims made in the paper lack rigorous support from the experimental results. The authors assert novelty in their approach to generalized additive models, yet prior literature already presents similar methodologies and much more impressive achievements. Specifically, the claim that this is the first paper to introduce such a model is incorrect—relevant prior work should be acknowledged (see references). Furthermore, the reported performance improvements are unconvincing. Based on results in Tables 1 and 2, when considering standard deviations, the proposed method performs comparably to existing approaches in both training and test settings. However, the baseline spline-based method achieves significantly smaller model sizes. This raises concerns about the practical significance of the proposed contribution, as the problem appears to be more effectively addressed by existing methods.

**Essential References Not Discussed:**

The paper omits several essential references related to generalized additive models and decision tree-based methods. Specifically, the authors claim to be the first to introduce a generalized additive model, but prior work has already developed such models with competitive performance and significantly fewer parameters. The following paper should be cited:

1. Fast Sparse Classification for Generalized Linear and Additive Models by Jiachang Liu, Chudi Zhong, Margo Seltzer, Cynthia Rudin, AISTATS 2022.

Additionally, the authors fail to compare their method against state-of-the-art decision tree-based models. The only comparisons provided are with basic tree models, which do not reflect the latest advancements in the field. The following papers present scalable and optimal sparse decision trees that should be considered:

2. Generalized and Scalable Optimal Sparse Decision Trees by Jimmy Lin, Chudi Zhong, Diane Hu, Cynthia Rudin, Margo Seltzer, ICML 2020.

3. Optimal Sparse Regression Trees by Rui Zhang, Rui Xin, Margo Seltzer, Cynthia Rudin, AAAI 2023.

These papers demonstrate that classification and regression trees can be constructed with fewer than 30 variables while maintaining performance comparable to much larger tree-based models developed using older methodologies. A thorough comparison with these methods is necessary to properly contextualize the contributions of this paper.

**Experimental Designs Or Analyses:**

The experimental design and analyses are standard practices in the ML community.

**Methods And Evaluation Criteria:**

The proposed alternating optimization method could lead to suboptimal solutions. In each iteration, the approach selects a feature and a corresponding split value while keeping the coefficients fixed. This rigid structure can lead to suboptimal feature and split choices, making the optimization process less refined. While such a method may offer some improvements over existing approaches, it lacks sophistication and does not provide guarantees on solution quality.

The evaluation criteria used in the paper align with standard machine learning practices, as they include training and test performance, model size, and training time. However, a critical issue is the absence of strong baseline comparisons. The experiments should include comparisons with state-of-the-art methods for both linear regression and graphical model selection to ensure a fair assessment of the proposed approach’s effectiveness.

**Other Comments Or Suggestions:**

Some experimental details, such as Lines 301–315 (right column), should be moved to the appendix. This section is overly detailed and does not contribute significantly to the main narrative. The main text should focus on the core contributions, while dataset descriptions and minor experimental details can be placed in supplementary material.

**Other Strengths And Weaknesses:**

The writing requires significant improvement. Many sections are overly verbose and lack substantive content. Certain passages could be condensed or moved to the appendix for better readability. For example, from lines 209 to 252, the authors spend nearly a full page deriving the objective function in Equation 5. However, this section reads more like a stream of consciousness than a structured derivation. Instead, the paper should present Equation 5 concisely at the beginning of the section, followed by a brief yet clear explanation.

Overall, the clarity and conciseness of the writing should be improved. The exposition is currently too wordy, making it difficult to follow the main contributions efficiently. Refining the explanations and eliminating redundancy would significantly enhance the readability and impact of the paper.

**Questions For Authors:**

1. How did you select the hyper parameters $\lambda$ and $\alpha$ in Equation 5, the objective function? To me, hand-picking the values does not seem like a rigorous approach of doing this.

2. Can you run [1, 2, 3], report results on datasets listed in Table 1 and 2, and make a comparison during rebuttal?

**Relation To Broader Scientific Literature:**

This paper is related to the broad research topics of interpretable machine learning, boosting, and decision trees. The key difference, compared with the boosting literature, is that historical stumps are allowed to be optimized over and over again.

**Theoretical Claims:**

There is no theoretical claim in this paper.

---

> ### Author Rebuttal · Authors · 2025-04-01
>
> Thanks for providing an insightful and constructive review! We greatly appreciate your effort in reviewing our paper.
>
> * **Claim on novelty.** To be precise, the only place we make a claim of doing something for the first time is with respect to ***learning stump forests with good generalization without ensembling techniques of boosting and bagging***.  At no point do we claim "to be the first to introduce a generalized additive model". Regarding reference [1], we acknowledge it as an important and relevant contribution. However, the main focus of [1] is not about decision stumps/trees but about learning sparse linear classifiers efficiently. The GAMs in [1] are introduced by learning a linear model on transformed features, where continuous variables are replaced with a set of binary indicators based on thresholds. We will include [1] in our literature review. We have already conducted empirical comparisons with [1] and updated our results table accordingly (see below).
>
> * **Unconvincing improvements compared to splines.** We acknowledge the concern regarding the magnitude of performance improvements. However, we would like to emphasize that GAMs, constrained to be sums of univariate functions, are inherently limited in their expressive power. Consequently, even an oracle, i.e., a globally optimal GAM, may yield only modest gains in predictive performance. Expecting substantial improvements over existing GAM variants may therefore be unrealistic given the inherent limitations of the model class.
>
>   On regression datasets, spline-based methods consistently underperform compared to stump- or tree-based approaches. As noted by Reviewer wGmH, a key advantage of piecewise constant GAMs over smooth methods like cubic splines is their ability to capture discontinuous shape functions and abrupt changes, which is an important characteristic for modeling tabular data. Another limitation of traditional splines lies in selecting the placement and number of knots (i.e., points where two spline segments meet). For example, PyGAM places the same number of knots for each feature, whereas in our method, the distribution of stumps is determined dynamically during optimization.
>
> * **Suboptimal solutions of alternating optimization.** To clarify, our algorithm performs *exact* optimization over the four parameters of one stump at a time: the feature index, threshold value, and the left and right leaf predictions. After a full pass through all stumps, all leaf predictions are globally optimized given the current split parameters. While the method may appear simple and may be perceived as lacking sophistication, it is in fact highly effective at minimizing error. For instance, as shown in the left plot of Figure 1, just 200 optimized stumps achieve the same error level as 1000 greedily added stumps.
>
> * **No theoretical claims or guarantees on solution quality.** As we elaborate in our rebuttal to reviewer wGmH, our algorithm has important theoretical properties: monotonic decrease of the objective at each iteration and convergence to a local optimum.
>
> * **Absence of strong baseline comparisons.** Our method operates within the class of GAMs, and as such our experimental comparisons include all the state-of-the-art methods (e.g. EBMs) within this model class. Other reviewers have acknowledged that the chosen baselines are appropriate and sufficient for evaluating our contributions within this context. In this rebuttal, we have also included an additional comparison with FastSparse.
>
> * **Additional comparisons.** We have run the methods from [1, 2, 3], and reported the results in Tables 1–4 of the rebuttal PDF: https://anonymous.4open.science/r/pdf_to_anon-1CD8/icml25_rebuttal_figures.pdf or https://drive.google.com/file/d/1Xso8hwFd9rB_XqAzH6H8ri7vUbrXEaq6/view?usp=sharing. Full experimental details are provided in the table captions. [1] generally delivers strong performance across many datasets while maintaining small model sizes and fast training times. We can also leverage the solution from [1] to warm-start our stump forests and achieve even more accurate results (see more in our rebuttal to owkV). In contrast, [2] and [3] were significantly slower to train; we had to apply subsampling and feature binarization to obtain usable results. While these tree-based models are interpretable, direct comparisons with them are outside the scope of our work, which focuses specifically on GAMs.
>
> * **Writing improvement.**  Thank you for the helpful suggestions regarding the writing. We will carefully incorporate these recommendations to improve clarity, conciseness, and overall readability in the final version of the paper.
>
> * **Selection of hyperparameters.** As we state in line numbers 555 to 563 of appendix, we perform grid-search over $\lambda = \{2.0, 4.0, 6.0\}$ for classification datasets, $\lambda = \{20.0, 40.0, 60.0\}$ for regression datasets. We use a fixed value of $\alpha = 0.1$, as it plays less significant role.

---

> > ### Comment · Reviewer_r73Z · 2025-04-04
> >
> > Thanks for the reply. I appreciate your running the additional experiments. I have several followup comments/questions.
> >
> > 1. When you optimize each individual stump, are you optimizing $\mu_l$ and $\mu_r$ as well or just optimizing over $\phi$ and $\tau$? If you are also optimizing over $\mu_l$ and $\mu_r$, how do you do that?
> >
> > 2. Thanks for running the FastSparse baseline. However, I find it hard to believe the reported results. Are you sure you are running the baseline correctly? If you look at Figure 8 in their AISTSTS paper, the authors were able to achieve prediction accuracy ~ 0.73 with only 15-20 coefficients. But in your rebuttal results, FastSparse uses 196 coefficients. The FastSparse paper also shows similar level of extreme sparsity it can achieve on GAMs for other two practical datasets. To me, FastSparse is either used or reported incorrectly in the rebuttal results.
> >
> > Yes, I know my criticism is much harsher compared to other two reviewers, but I work a lot on $\ell_0$ optimization, and I am kind of sure $\ell_0$ optimization can produce much sparser models than $ell_1$ optimization or the results shown in this paper. Thus, I find it hard to reconcile this sentence "Our regularized stump forests achieve accuracy comparable to state-of-the-art GAM methods while using fewer parameters" stated in the abstract. The reason is that there are already $\ell_0$ optimization methods that can achieve much much sparser GAM models.
> >
> > If you really want to claim you can achieve much sparser models, I recommend you do some plotting similar to Figure 5 and Figure 8 shown in the FastSparse AISTATS paper. Specifically, plot accuracy (or AUC or squared error) vs sparsity level on both the training and test sets.
> >
> > 3. For the two decision tree baselines, when you report "out of running time", do these baselines not even return any tree at all? Usually for methods based on branch-and-bound (BnB), if they exceed the running time and optimality is not certified, many BnBs just return the best heuristic solution found so far. So in principle you can at least use the best heuristic model to do evaluation on the training and test sets.

---

> > > ### Author Response · Authors · 2025-04-09
> > >
> > > * We enumerate all possible split candidates $(\phi, \tau)$. For each potential split, the optimization over $\mu_l$ and $\mu_r$ is strongly convex and can be solved exactly. In the regression setting (we show the case where $\alpha = 0$ due to space constraints), the problem is: $\min_{\mu_l, \mu_r} \frac{1}{2} \sum_{n \in l} (y_n - \mu_l)^2 + \frac{1}{2} \sum_{n \in r} (y_n - \mu_r)^2  + \lambda \ |\mu_l - \mu_r|$. Let $\bar{y}_l$ and $\bar{y}_r$ denote the sample means of the left and right partitions, and let $n_l$ and $n_r$ be the number of points in each. The optimal values of $\mu_l, \mu_r$ admit a closed form solution:
> > > $$
> > > \mu_l = \bar{y}_l - \frac{\lambda}{n_l}, \ \mu_r = \bar{y}_r + \frac{\lambda}{n_r} \quad \text{if  } \bar{y}_l - \bar{y}_r > \lambda (\frac{1}{n_l} + \frac{1}{n_l})
> > > $$
> > > $$
> > > \mu_l = \bar{y}_l + \frac{\lambda}{n_l}, \ \mu_r = \bar{y}_r - \frac{\lambda}{n_r} \quad \text{if  } \bar{y}_l - \bar{y}_r < -\lambda (\frac{1}{n_l} + \frac{1}{n_l})
> > > $$
> > > $$
> > > \mu_l = \mu_r  = \frac{n_l \bar{y}_l + n_r \bar{y}_r}{n_l + n_r} \quad \text{otherwise}.
> > > $$
> > > By maintaining cumulative statistics when scanning over split points, we can evaluate the optimal $\mu_l, \mu_r$ and the corresponding objective for each split in constant time. This gives an overall linear-time procedure, assuming the features are pre-sorted.
> > >
> > >   For classification with cross-entropy loss, the optimal values of $\mu_l, \mu_r$ do not admit closed-form solutions. Instead, we approximate them using a single Newton step, an established technique used in XGBoost and in the iteratively reweighted least squares (IRLS) method for traditional GAM backfitting. Given that our framework revisits stumps at each iteration and jointly optimizes all leaves, these Newton updates are both computationally efficient and well-aligned with the structure of our iterative optimization procedure.
> > >
> > > * In our experiments, the reported model size for piecewise constant GAMs is defined as the number of constant segments *multiplied by 2*, to account for both the split threshold and the constant value.
> > >
> > >   For all baselines, we follow the same protocol: hyperparameters are selected using grid search on a validation set, and final performance is reported as the average over five independent train/test splits using the best-found parameters. For FastSparse we use the built-in num_lambda=100 setting to search for the optimal value of $\lambda$ during hyperparameter tuning.
> > >
> > >   One implementation detail is that the FastSparse code does not provide functionality for learning a feature binarization scheme from the training data and applying it to the test data. Based on their paper and available code, it seems that feature binarization may have been performed on the entire dataset prior to model training, which can lead to information leakage from test to training data. In contrast, we ensure that binarization thresholds are obtained strictly from the training set and then applied to the test set.
> > >
> > >   We observe inconsistency in FastSparse’s behavior depending on whether a given $\lambda$ model is part of a regularization path. For example, training a single model with $\lambda = 0.63$ (by specifying the lambda_grid argument with a single value) results in 102 nnz coefficients, while the same $\lambda$ within a regularization path produces 71 nnz coefficients, despite all other parameters being equal. This sensitivity does not seem to be documented in the FastSparse paper or codebase. Because FastSparse seems to favor the use of regularization paths, we compare its full path to that of our method (see below).
> > >
> > >   We fully agree that $\ell_0$-based optimization is a more effective approach for obtaining sparse models compared to $\ell_1$ regularization. This is, in fact, **a key reason why our method produces compact models: we explicitly limit the number of stumps from the outset, effectively imposing an $\ell_0$-type constraint**, and then optimize over this restricted stump budget. The $\ell_1$-roughness penalty in our approach is primarily intended to control overfitting. We do not set its value high enough to enforce sparsity; rather, we rely on it for its shrinkage effect.
> > >
> > >   We perform experiments comparing full regularization paths, and produce the suggested figs. 1 and 2 in <https://anonymous.4open.science/r/pdf_to_anon-1CD8/icml25_rebuttal_figures_2.pdf> or <https://drive.google.com/file/d/1WZpBhmgvDD0QYV5-X-1aKuchCgRa_TdK/view?usp=sharing>. The figure captions provide descriptions of the experiments. On the FICO dataset, our method performs comparably to FastSparse. On the cpuact dataset, our approach achieves higher accuracy, which we attribute to the regularizing effect of the $\ell_1$ roughness penalty.
> > >
> > > * When the time limit is reached, the GOSDT implementation terminates with an exception: "Error: GOSDT encountered an error while training". The OSRT implementation produces some models if it exceeds the time limit, and we have updated our results table with it.

---

### Official Review · Reviewer_wGmH · 2025-03-24

**Overall Recommendation:** 4

**Summary:**

The paper introduces a method to fit decision tree stumps per feature, which can be interpreted as a generalized additive model. To mitigate overfitting, the authors propose a smoothness constraint that is optimized jointly with the stump parameters. Unlike approaches such as EBMs that greedily add decision tree stumps, this method requires fewer stumps because it globally optimizes stump placement according to the data. Consequently, the method avoids the need for ensemble strategies like bagging and boosting. The experiments demonstrate comparable performance to Explainable Boosting Machines and other baselines, but using a more straightforward fitting procedure.

**Claims And Evidence:**

Yes, all claims of the paper are well supported.

**Essential References Not Discussed:**

I am not aware of any missed references.

**Experimental Designs Or Analyses:**

The experimental design seems solid. The hyperparameters of each baseline are optimized using a grid search over several hyperparameters. ORSF performs well throughout

**Methods And Evaluation Criteria:**

Yes, all considered benchmarks are classic benchmarks to consider in the GAM literature.

**Other Comments Or Suggestions:**

-

**Other Strengths And Weaknesses:**

### Positive:
- The framing of learning the decision tree stumps using a global optimization approach rather than a greedy method is well motivated. The problem with overfitting is illustrated well and the proposed regularizer is intuitive to understand.
- Unlike Neural Additive Models the proposed approach can easily model discontinuities in the shape functions like EBMs which NAMs struggle to. I think the authors could emphasize this even more because discontinuities can often occur in tabular data and the GAM should be able to represent this.
- The paper is well written and easy to understand.


### Weaknesses:
- Could you provide a theoretical justification or discussion about whether the proposed alternating optimization approach can reliably reach a global optimum?
- Could you also show the shape functions of the other methods for comparison?
- You might consider incorporating pairwise interactions by applying the same strategy used by EBMs to identify dominant pairwise effects (Lou et al., 2013). Using similar pairwise effects would allow an additional comparison point with EBMs.

### References:
- Lou, Y., Caruana, R., Gehrke, J., & Hooker, G. (2013, August). Accurate intelligible models with pairwise interactions. In Proceedings of the 19th ACM SIGKDD international conference on Knowledge discovery and data mining (pp. 623-631).

**Questions For Authors:**

My understanding is that your method implicitly places more stumps in regions with higher data density, which is typically the case. However, Figure 5 (feature "year") shows a surprisingly uniform distribution of thresholds. Could you clarify why this occurs?

## update after rebuttal
- The authors addressed all of my questions and looking at the other two reviews I have no further questions.

**Relation To Broader Scientific Literature:**

This paper proposes a new method of fitting decision tree stumps, which is not a novel model class to be used in GAMs as they are also used prominently in EBMs. However, the fitting procedure discussed by the authors is novel as far as I know. It's intuitive as it globally optimizes the loss function.

**Theoretical Claims:**

-

---

> ### Author Rebuttal · Authors · 2025-04-01
>
> Thank you for providing us with a very valuable and insightful review! We greatly appreciate your effort in reviewing our paper.
>
> * **Modeling discontinuities.** You are absolutely right that NAMs (as well as traditional cubic splines) are limited in their ability to capture sharp discontinuities in shape functions. This is a significant limitation in many real-world tabular datasets, where such discontinuities are common due to thresholds, categorical splits, or other sharp nonlinearities in the data. One of the key advantages of the stump-based shape functions, similar to those used in EBMs, is their natural ability to represent abrupt changes locally in the function without inducing unnecessary smoothness. This is also a likely reason why NAMs tend to underperform in our experiments on tabular datasets. We really appreciate your pointing this out, we will incorporate this discussion more explicitly in the introduction to better highlight this advantage of our method.
>
> * **Reaching a global optimum** Although we do not formally prove this result, and cannot easily find a definitive reference establishing it, the problem of learning an optimal decision stump forest appears to be computationally intractable in general. Intuitively, it resembles the well-known best subset selection problem, which is NP-hard. Each decision stump is specified by a feature index and a threshold. However, selecting the best set of $T$ stumps entails choosing an optimal subset of $T$ feature-threshold pairs from a combinatorially large candidate space. For a dataset with $N$ samples and $D$ features, there are $O(ND)$ such candidates, leading to a total search space of size $\binom{ND}{T}$. This exponential growth in possibilities suggests that the overall problem may be NP-hard, implying that finding a globally optimal solution in polynomial time is unlikely in the general case.
>
> * **Theoretical properties.** Nevertheless, our proposed alternating optimization algorithm has important theoretical properties. Specifically, each optimization step is guaranteed to monotonically decrease the overall objective, and the algorithm converges to a local optimum where no single alternating optimization step can further improve the solution (quite similar to the k-means clustering algorithm). The traditional ensembling methods of boosting and bagging do not provide these type of theoretical guarantees, and lack a well-defined global objective, unlike our more principled optimization framework. The ability of our approach to take an initial stump forest, and optimize it further has the additional advantage of warm-starting behavior: the model can be initialized from an existing forest, such as one derived from any piecewise constant GAM (more on this in our rebuttal to reviewer owkV), and further refined. This also allows for efficient model updates when new training data becomes available (more on this in our rebuttal to reviewer owkV).
>
> * **Shape functions of the other methods.** In https://anonymous.4open.science/r/pdf_to_anon-1CD8/icml25_rebuttal_figures.pdf or https://drive.google.com/file/d/1Xso8hwFd9rB_XqAzH6H8ri7vUbrXEaq6/view?usp=sharing, in figs. 2 and 3, we provide the shape functions for EBM and PyGAM. Overall, the general trend in EBM shapes is quite similar to the ones obtained by our method and also being similar in accuracy (around 2,200 test RMSE), although it generated an overly noisy curve for the mileage feature. PyGAM, on the other hand, generates very smooth curves, and behaves quite unpredictably in regions with less or no data.
>
> * **Pairwise interactions.**  (Lou et al., 2013) uses a two-step approach for building GA$^2$M: first, building univariate shapes, then modeling and including pairwise interactions on the residuals using a variation of depth-2 decision trees. Indeed, we can directly apply a similar strategy, and extend our method to model pairwise interactions. But we believe that it is possible to extend the proposed alternating optimization method to learn an ensemble of both decision stumps and depth-2 trees. Naive implementation of this idea could result in overfitting, and the regularization terms of our current approach is not directly applicable to depth-2 trees. Defining effective regularization techniques for these more complex tree ensembles can be an important future research direction.
>
> * **Distribution of thresholds on the "year" feature.** The "year" feature is highly discretized, with only 22 unique values ranging from 1999 to 2020. In the higher-density range (2013–2020), nearly all possible thresholds are utilized. In contrast, some thresholds are skipped in lower-density regions—for example, 2005 and 2008 are not used. While decision stumps may appear to concentrate in regions with more data, their distribution is ultimately determined by the optimization process rather than simply by feature density.

---

### Decision · Program_Chairs · 2025-05-01

**Decision:**

Accept (poster)

**Comment:**

The paper proposes a new method for fitting generalized additive models using decision stumps, by jointly optimizing all model parameters. This results in much smaller models than alternative methods such as EBMs, while providing competitive predictive performance. Additional improvements and experiments during the rebuttal period show runtime similar and often faster than for EBMs.
This is an interesting new angle on fitting piecewise constant GAMs, which compares favorably with existing methods.